# Massive Irreparable Rotator Cuff Tears: Which Patients Will Benefit from Physiotherapy Exercise Programs? A Narrative Review

**DOI:** 10.3390/ijerph20075242

**Published:** 2023-03-23

**Authors:** Eoin Ó Conaire, Ruth Delaney, Alexandre Lädermann, Ariane Schwank, Filip Struyf

**Affiliations:** 1Department of Rehabilitation Sciences and Physiotherapy/MOVANT, Faculty of Medicine and Health Sciences, University of Antwerp, Universiteitsplein 1, 2610 Wilrijk, Belgium; ariane_schwank@hotmail.com (A.S.); filip.struyf@uantwerpen.be (F.S.); 2Evidence-Based Therapy Centre, First Floor Geata na Cathrach, Fairgreen Road, H91 W26K Galway, Ireland; 3Dublin Shoulder Institute, Sports Surgery Clinic, Santry, D09 C523 Dublin, Ireland; ruth@dublinshoulder.com; 4Division of Orthopaedics and Trauma Surgery, La Tour Hospital, 1217 Meyrin, Switzerland; alexandre.laedermann@gmail.com; 5Faculty of Medicine, University of Geneva, 1205 Geneva, Switzerland; 6Division of Orthopaedics and Trauma Surgery, Department of Surgery, Geneva University Hospitals, 1205 Geneva, Switzerland; 7Institute for Therapy and Rehabilitation, Canton Hospital Winterthur, 8400 Winterthur, Switzerland

**Keywords:** review, massive, rotator cuff, irreparable, conservative treatment, non-operative, physiotherapy, rehabilitation, prognosis, prognostic

## Abstract

Massive irreparable rotator cuff tears can cause significant shoulder pain, disability and reduction in quality of life. Treatment approaches can be operative or non-operative. Operative approaches include reverse total shoulder arthroplasty, arthroscopic debridement, partial rotator cuff repair, subacromial balloon spacers, superior capsule reconstruction, and tendon transfer procedures. Non-operative approaches include physiotherapy exercise programs and corticosteroid injections. There are no randomized controlled trials comparing the different treatment approaches. It is therefore challenging for clinicians to advise patients on what is their best treatment pathway. Physiotherapy exercise programs are less expensive and have lower risks for patients than surgical approaches. However, the success of physiotherapy in patients with massive irreparable rotator cuff tears is highly variable with published success rates of 32–96%. Several cohort studies have sought to identify if certain factors are predictive of success with physiotherapy. Several biomechanical factors were identified as possibly being related to a successful or unsuccessful outcome following physiotherapy, with complete tear of subscapularis demonstrating the strongest evidence. However, there were no appropriately designed prognostic studies. There has been a strong emphasis on biomechanical factors. Other domains such as psychosocial factors, which are important in similar patient populations, have not been explored. We recommend that further research is needed in this area and should include randomized controlled trials comparing treatment approaches and longitudinal prospective prognostic studies to identify predictors of treatment success.

## 1. Introduction

### 1.1. The Patient

Rotator cuff-related shoulder pain which refers to the spectrum of shoulder conditions including, subacromial pain (impingement) syndrome, rotator cuff tendinopathy, and symptomatic partial and full thickness rotator cuff tears, is the most common cause of shoulder pain [1,2,3]. One sub-group, massive irreparable rotator cuff tears (MIRCT), is associated with significant disability. While some patients can be asymptomatic, many experience worsening pain and weakness in the involved shoulder and pain when they elevate the affected arm above shoulder height or in some cases above 45° [4,5]. This can have a profoundly negative impact on quality of life with pain and difficulty during many basic everyday activities, such as washing, dressing, sleeping, housework, food preparation, overhead activities and work [6].

### 1.2. Biomechanics

Normal shoulder kinematics are dependent on the head of the humerus being centred in the glenoid by the synergistic action of the rotator cuff and deltoid muscles. In cases of MIRCT, disruption of the activity of the rotator cuff can lead to unopposed action of the deltoid resulting in superior migration or superior and anterior subluxation [7]. This uncontrolled movement of the humeral head can lead to it abutting the coracoacromial arch, thus leading to a mechanical block to motion and a reduction in active movement [8]. When this reduction in active movement occurs, but full passive range of motion is preserved, it is called pseudoparesis or pseudoparalysis [9]. Intriguingly, many patients with MIRCT can achieve full range of motion, apparently by maintaining balanced force couples in the glenohumeral joint [4]. This is thought to be due to the presence of the anterior and posterior rotator cables. These thickened portions of the anterior and posterior rotator cuff attachments onto the humerus allow the remaining muscle fibres to transmit their forces across the glenohumeral joint [10].

### 1.3. Surgical Treatment Approaches

All treatment approaches aim to regain active motion above shoulder level, reduce or remove pain and restore good function to the affected arm [11]. Historically, Reverse Total Shoulder Arthroplasty (RTSA) was seen as a promising treatment option for patients with MIRCT with advanced osteoarthritis. RTSA was considered a salvage operation for older patients with advanced arthropathy, pseudoparalysis, and very low levels of function [12]. RTSA involves implanting a large glenoid hemisphere that articulates with a humeral cup, moving the centre of rotation more medially and distally, preventing superior migration of the humerus and making it biomechanically easier to elevate the arm above shoulder height [13]. More recently, with improvements in implant design and surgical techniques, RTSA has demonstrated positive short and mid-term results in patients with MIRCT without glenohumeral arthritis [14,15]. However, these positive results are tempered by concerns about reported complication rates of 15–20% or up to 39% in some cohorts [14,15,16]. The operation remains controversial for younger patients without glenohumeral osteoarthritis. Consequently, a wide range of other surgical approaches for this patient group ranging from arthroscopic debridement, partial rotator cuff repair, subacromial balloon spacers, superior capsule reconstruction, and tendon transfer procedures, has been developed [17].

### 1.4. Non-Operative Treatment Approach

Many patients with MIRCT have successful outcomes with non-operative management. By its nature, physiotherapy is less costly and less risky than surgery, but the overall outcomes are very variable. Some patients appear to achieve successful outcomes, while some do not. A recent systematic review of physical therapy for patients with MIRCT demonstrated outcomes ranging from 32% to 96% overall success for functional outcome, strength, and range of motion [18]. Physical therapy for MIRCT primarily involves exercises that aim to correct the uncontrolled movement of the humeral head from the glenoid and thus generate improvements in range of motion, strength, function, and quality of life. Physiotherapy can also focus on scapular muscle rehabilitation, proprioceptive training, and relief of stiffness if present [5,18]. There is uncertainty about what constitutes the most effective physiotherapy program.

### 1.5. Clinical Decision-Making

On a day-to-day basis, the difficulty for clinicians is to advise individual patients accurately and confidently on their best treatment pathway. There are no randomized controlled trials comparing the treatment approaches, and there is no clear consensus on what intervention is superior [19]. This clinical therapeutic challenge is reflected in one case-based survey of specialist shoulder surgeons, demonstrating only fair agreement regarding treatment selection for this patient group [20]. This has led to a situation where the same patient may be offered physiotherapy, partial repair, tendon transfer, superior capsule reconstruction, subacromial balloon spacer, or RTSA depending on factors such as where the person lives, the surgeon’s experience, or the cost of treatment [7]. Therefore, rather than stratified healthcare or personalized medicine, patients with MIRCT are typically offered “stepped” models of care with a sequence of interventions that increase in invasiveness, beginning with a six-month trial of physiotherapy [2].

### 1.6. Aim

The primary aim of this narrative review was thus to identify possible prognostic factors for both successful and unsuccessful outcomes from physiotherapy in patients with MIRCT. To facilitate this primary objective, we comprehensively reviewed the literature related to MIRCT and critically evaluated the theory, content, parameters and outcomes of different physiotherapy programs.

### 1.7. Method

A systematic review methodology was originally initiated, and Cochrane, Cinahl, Medline, PubMed and PEDro databases were searched. The keywords were derived from the research question and from reviewing recent relevant articles on the topic. The search strategy included the following keywords: “shoulder”, “glenohumeral”, “massive rotator cuff tear”, “massive irreparable rotator cuff tear”, “complete rotator cuff tear”, “rotator cuff tear”, “physiotherapy”, “rehabilitation”, “non-operative management”, “conservative management”, “exercise”, “prognosis”, “prognostic”, “predictors”. Exclusions included non-English language papers, protocols in trial registries that had not been published, and studies including populations without MIRCT. Screening of the results based on the titles and abstracts revealed that there were no prospective prognostic cohort studies and no randomised controlled trials fitting our definition of MIRCT. We concluded that a systematic review methodology or meta-analysis was not possible. However, as Greenhalgh et al. [21] have pointed out, “the narrative review structure selects evidence judiciously and purposively with an eye to what is relevant”. We made the decision to change the methodology to a narrative review and therefore, broadened our search strategy to include case series studies of patients with massive rotator cuff tears and studies identifying prognostic factors from other related shoulder presentations, adding the search terms “rotator cuff related shoulder pain”, “rotator cuff tendinopathy”, “subacromial impingement” and “shoulder pain”. We also hand-searched the reference lists of relevant articles. We believe that this has provided a more nuanced and clinically applicable review to collate and extend knowledge and set research priorities in the area.

## 2. Definitions

### 2.1. Massive Rotator Cuff Tear

Historically, rotator cuff tears exceeding 5 cm were described as massive rotator cuff tears [22]. However, this definition does not consider the patient’s height and, therefore, the size of the involved tendon. More recently, a more widely accepted definition reserves the term “massive tear” for when two or more rotator cuff tendons are torn with at least one of the two tendons retracted beyond the top of the humeral head [23,24]. Massive rotator cuff tears make up approximately 10–40% of all rotator cuff tears [25] and 80% of all recurrent tears [26,27].

### 2.2. Irreparable Rotator Cuff Tears

Not all massive rotator cuff tears are considered irreparable. Because massive tears often occur in an elderly population with poor quality, fragile tendon tissue attempted surgical repair of the tendons is associated with high failure rates. A review of 18 studies reporting outcomes after repair of massive rotator cuff tears found a retear rate of 78% [28]. Because of this high failure rate, several ways of identifying which massive rotator cuff tears are likely to be irreparable have been developed.

### 2.3. Ways to Identify Massive Irreparable Rotator Cuff Tears

Tendon retraction

Patte et al. [24] devised a classification based on the amount that a torn rotator cuff tendon has retracted from its insertion on the greater tuberosity. Grade 1 relates to little tendon retraction, Grade 2 to retraction of tendon to the level of the humeral head, and Grade 3 describes retraction of the tendon to the level of the glenoid (Figure 1).

2.Fatty infiltration

Goutallier et al. [29] introduced a classification system based on the amount of fatty infiltration in the rotator cuff muscles. Stage 0 indicates no fatty degeneration; stage 1 has some fatty streaks; stage 2 has less fat than muscle; stage 3 has as much fat as muscle, and stage 4 has more fat than muscle (Figure 2). Several studies have shown that tears involving more than one tendon with Goutallier Grade 3 or 4 fatty infiltration and significant tendon retraction (Patte Grade 3) are much less likely to have success with a surgical repair [30,31,32]. The subgroup of tears with these radiological features is commonly called MIRCT.

However, the terminology has drawn some controversy and debate as many orthopaedic surgeons have stated that the only way to truly determine if a tear is irreparable is for the patient to undergo arthroscopic examination and attempted repair [31]. While this may be strictly speaking correct, the use of predictive factors, such as the Goutallier and Patte criteria, have strong correlation with irreparability.

3.Acromiohumeral distance

In addition, an acromiohumeral distance of less than 7 mm due to superior migration of the head of the humerus is associated with irreparability [33] (Figure 3).

4.Positive tangent sign

A positive tangent sign has also been associated with irreparability [33]. This is a radiological sign on MRI scan where a line is drawn through the superior borders of the scapular spine and the superior margin of the coracoid [34]. The tangent sign is considered present when the supraspinatus muscle does not cross the tangent indicating supraspinatus atrophy (Figure 4).

It is neither practical nor helpful to reserve the term irreparable to describe only those patients who have had surgical exploration. Therefore, throughout this review, the term MIRCT will be applied to patients with a complete rupture of two or more tendons, with retraction from the tendon insertion and Goutallier Grade 3 or 4 fatty infiltration of the affected muscles.

### 2.4. Pseudoparesis versus Pseudoparalysis

The terms pseudoparesis and pseudoparalysis are used interchangeably and inconsistently in the literature related to MIRCT [8,9,35]. Some authors define pseudoparalysis as a massive rotator cuff tear that leads to a limited active forward flexion of less than or equal to 45° without passive restriction or neurological deficit, and that pseudoparesis differs only by having limited active forward flexion of greater than 45° and less than or equal to 90° [35]. However, in one systematic review on the topic the authors highlighted that the most common definition of pseudoparalysis in the literature was active elevation of less than 90° with full passive elevation [9]. These authors attempted to gain clarity on the terminology by defining pseudoparalysis as a massive rotator cuff tear with 0° of active elevation and full passive elevation, and pseudoparesis as a massive rotator cuff tear with less than 90° active elevation and full passive elevation. The lack of agreement on terminology was highlighted in another systematic review which identified that of the 16 studies included, four did not differentiate between pseudoparalysis and pseudoparesis, five studies did not define either, and seven studies had heterogenous definitions [8]. This lack of consensus is further highlighted by a survey of 246 shoulder surgeons which demonstrated a lack of agreement on how best to define pseudoparalysis, and low inter-rater agreement when assessing via a video consultation [36]. It would appear that despite these efforts to gain clarity on the definitions of these terms, there is still widespread lack of agreement.

## 3. What Constitutes the Best Physiotherapy Program?

Before considering which factors may be related to a successful or unsuccessful outcome from physiotherapy, it is essential to consider the structure, content, and parameters of the physiotherapy programs for MIRCT that have been evaluated in the literature. A variety of exercise therapy interventions have been used.

Many authors have used an “anterior deltoid strengthening program” [37,38,39,40]. The rationale for this type of rehabilitation program is based on the suggestion that strengthening the anterior deltoid can compensate for superior migration of the head of the humerus and consequently allow improved congruence of the humeral head in the glenoid, resulting in an improved ability to actively raise the arm [39]. The four studies that have used this rehabilitation approach demonstrated good, but variable success rates (40–100%) in this patient group. However, the rationale has been questioned. First, the assertion that these exercise programs—which focus primarily on re-training shoulder flexion—specifically target the anterior deltoid has not been confirmed by research. Second, despite significant improvements in range of motion, strength, patient-reported function, and quality of life, no studies demonstrate increased cross-sectional area of anterior deltoid or changes in anterior deltoid muscle activity levels following the program [38]. Third, it has been suggested that this approach could theoretically cause anterior translation of the humeral head and *reduce* congruency of the glenohumeral joint during movement [41]. Collin et al. describe an alternative rehabilitation approach targeting the entire deltoid and the muscles that stabilize the scapula and focusing on exercises in an elevated position [7]. The rationale is that the entire deltoid is not working to elevate the arm in this position, but rather acts synergistically with the remaining rotator cuff muscles to enhance the centring of the humeral head in the glenoid. The program also focuses on relieving muscle tension in the scapular and neck areas, gentle manual techniques, scapular muscle re-education to optimize glenoid position, strengthening the remaining rotator cuff, and proprioceptive and neuro-motor rehabilitation with an emphasis on visual cues.

Another case series study provided a detailed description of the physiotherapy intervention, which consisted of two joint-based manual therapy techniques and exercises [42]. Their approach followed a consensus statement, which recommended that physiotherapy treatment decisions be based on the clinical assessment findings rather than on structural pathology [43]. Many authors [41,44] exclude patients with passive restriction of shoulder movement. Applying a broader physiotherapy program adapted to each patient’s clinical findings irrespective of their diagnosis may mean that more patients could gain improvements.

The delivery of exercise interventions is highly variable across the available studies. Some studies used therapist-supervised exercise sessions [42,45], while others used home exercise programs [39,40]. The duration of the exercise intervention also varied. Some studies reported that the exercise program was carried out for up to twelve weeks [39,40,42], whereas others reported continuing exercise for up to five months [38] or even up to twelve months [44]. Other investigators did not state a specific timeframe, but instead referred to the number of physiotherapy sessions [45].

The variation in success of non-operative treatment may be due to this variability in the delivery, duration, and content of the rehabilitation programs. Following their recent systematic review of non-operative therapy for massive rotator cuff tears, Shepet et al. [18] developed a synthesized non-operative treatment protocol [18]. They acknowledged that this was challenging because several studies that showed improvements with exercise therapy did not report their treatment protocols and so could not be included. Therefore, their suggested standardized rehabilitation program is based on just four studies [37,38,39,42]. The authors acknowledge that this protocol is based on low-level evidence. However, it provides one starting point for future randomized controlled trials as a comparator treatment. In addition, it identifies key elements of the delivery of the rehabilitation which the studies with the best outcomes had in common. These key elements included manual therapy, passive range of motion exercises, postural, scapular stabilization and proprioceptive exercises and a structured progressive strengthening program. It was proposed that pain should be monitored closely when carrying out the exercises and should be no more than 4/10 on the visual analogue scale during or after the exercises [18]. There has been considerable debate in the scientific literature about whether exercises for rotator cuff related shoulder pain should be carried out symptom-free [46] or in the presence of pain [47], but no studies have yet evaluated this in patients with MIRCT.

Overall, it would appear from the literature that exercise programs that emphasize improving shoulder flexion have some efficacy. However, there is no clear evidence that these exercise programs specifically or preferentially target the anterior deltoid. In fact, EMG studies have demonstrated that shoulder flexion exercises preferentially activate supraspinatus and infraspinatus and that shoulder extension exercises preferentially activate subscapularis [48]. Other physiotherapy approaches have also demonstrated efficacy in this patient group, but there are no comparative randomized controlled trials to evaluate the benefits of one approach over another. It is also clear from the literature that research is lacking on how successful different physiotherapy programs are at achieving their intended goal of centring the head of the humerus in the glenoid. We are not aware of any studies that have evaluated radiological signs of congruence—such as acromiohumeral distance—in patients who have successfully completed physiotherapy. Finally, despite the likelihood that patient expectations, therapeutic alliance, and engagement with the physiotherapy program are important factors associated with a successful or unsuccessful response to physiotherapy [49,50], we are not aware of any reported physiotherapy programs for this patient group that either measure therapeutic alliance or consider approaches which might moderate patient expectations.

## 4. How Successful Are Physiotherapy Exercise Programs in Patients with MIRCT?

Physiotherapy has been shown to have variable success in people with massive irreparable rotator cuff tears. Comparing the results of two recent systematic reviews reveals conflicting interpretations of the literature. Firstly, the above-cited systematic review [18] reported marked variability of outcomes (32–96% overall success). The review reported that all ten included studies were of low-quality evidence (Level III and IV) with a high risk of bias. Another systematic review evaluated physiotherapy and surgical outcomes [51]. The authors also acknowledged the low-quality evidence (Level III and IV) and only included three studies related to physiotherapy. They excluded studies included in the Shepet et al. [18] review because other tear sizes were included in the cohort; the follow-up time was deemed too short, or the outcome measurements were insufficient. When they pooled the data across the studies, the authors concluded that 60% of patients did not respond to physiotherapy or went on to have surgery, and that physiotherapy compared to surgery may lead to high failure rates and inferior outcomes. It is interesting to note that with this different approach to study selection for inclusion in the review, a more negative picture was painted of this patient group’s response to physiotherapy.

When we examine individual studies, we can see the variability in successful outcomes in this group of patients. Review of the literature revealed no randomized controlled trials evaluating response to physiotherapy in patients with MIRCT. One prospective case series study assessed the response of an elderly cohort (*n* = 17) with irreparable rotator cuff tears to an anterior deltoid re-education program [39]. The authors demonstrated a success rate of 82% at nine months. However, success is defined as “adequate range of motion” and is therefore somewhat vague. A subsequent case series (*n* = 30) with similar methodology could not replicate these results, demonstrating a success rate of only 40% at two years [40]. In this study, the definition of a successful outcome was clearer. Patients were deemed to have failed if they decided to proceed to have surgery or abandoned the exercise program due to pain or lack of progress. Both studies had very clear inclusion criteria for MIRCT: complete rupture ≥ 2 tendons, Goutallier Grade 4 fatty infiltration, Patte Grade 3 retraction, and described similar flexion-based exercise programs for at least twelve weeks. The difference in outcome success of the two similar studies may lay in the definition of outcome satisfaction. The outcome “adequate range of motion” may not sufficiently represent the facets of pain, reasons to abandon the exercise program or lack of progress. Neither study included a holistic patient-centred view incorporating outcome measures, such as patients’ expectations, perceptions, pain and quality of life.

Another case series of patients (*n* = 24) with MIRCT evaluated a longer duration exercise protocol (five months). The results demonstrated improved function in the symptomatic shoulder, reduced pain, and increased quality of life [38]. Gutierrez-Espinoza et al. (2018), in their case series of patients with MIRCT, demonstrated statistically significant short-term improvements in pain and function after twelve weeks of physiotherapy [42]. A subsequent paper reported the 12-month follow-up results of this cohort of patients demonstrating that the improvements were maintained at one year [52]. Both studies reported statistically significant improvements, but did not define or evaluate what would be considered a clinically meaningful improvement. Table 1 summarises the characteristics and findings of the studies evaluating physiotherapy for patients with MIRCT.

## 5. What Are the Predictors of a Successful or Unsuccessful Response to Physiotherapy in Patients with MIRCT?

Because of the variability of response to physiotherapy in patients with MIRCT, several studies have attempted to identify factors that may predict a successful or unsuccessful response to rehabilitation. One case series found that shoulder flexion of less than 50° at the outset of the program was associated with an unsuccessful result from physiotherapy [40]. Some authors have hypothesized that the location (which tendons are involved) of the tear may be an important factor associated with the outcome [7,45]. One innovatively designed case series of patients with MIRCT (*n* = 45) tested this hypothesis [45]. All study participants had an active range of motion of less than 90° and full passive range of motion and the authors defined a successful outcome as patients achieving 160° active forward flexion. Following five sessions of physiotherapy, 25/45 (55%) of the patients had a successful outcome. Interestingly, of those patients with massive tears of the posterior rotator cuff (supraspinatus and infraspinatus), 14/15 (93%) had a successful outcome and recorded substantial improvements in range of motion. Conversely, those patients with tears involving subscapularis had much poorer outcomes, with only 6/21 (29%) having a successful outcome. Additionally, in those with tears of three tendons, only 7/16 (44%) had successful outcomes. Notably, the rehabilitation program employed was novel in that it did not focus on anterior deltoid strengthening, but rather on strengthening the scapular muscles and the entire deltoid muscle. Although the participant numbers in the study are relatively small for a prognostic study, and the methodology is not designed to determine the influence of other possible prognostic factors, it identifies a likely association between the site of the tear and response to physiotherapy.

The critical role of subscapularis in stabilizing the humeral head in the glenoid has a solid theoretical basis. It is substantially larger than any of the other rotator cuff muscles, and its force-generating capacity has been estimated to be equal to the other three rotator cuff muscles combined [53]. In many ways, it acts as two separate muscle components and with different electromyographic activity between the superior and inferior portions [54]. The superior part (innervated by the superior subscapular nerve) makes up two-thirds of the muscle and narrows to a tendon at the lesser tuberosity, whereas the inferior part (innervated by the lower subscapular nerve) remains muscular in its attachment. As precursors to their study on response to physiotherapy in this patient group, Collin et al. have written insightfully about functional deficits with different rotator cuff tear patterns [41]. Their study demonstrated that 80% of patients with complete supraspinatus, infraspinatus and subscapularis tears were unable to actively elevate the arm above 90° (and had preserved passive range of motion) compared to 45% of patients with complete tears of supraspinatus, infraspinatus and superior subscapularis. In another case series of 32 patients, the same authors found that patients could not actively raise the arm above 90° in 80% of cases when the inferior part of the subscapularis was torn, but never when only the superior part was torn [55].

In another cohort study (without a control group or randomization to the treatment group), the investigators found that patients with an intact subscapularis tendon or teres minor hypertrophy had a 57% rate of success with non-operative treatment compared to 32% in patients who had complete rupture of subscapularis or lacking teres minor hypertrophy [56]. However, another prospective cohort study (*n* = 71) did not find any correlation between the location of the tendon rupture and the success of non-operative treatment [44]. It is difficult to draw firm conclusions from this study as only 34/71 participants underwent physiotherapy. The remainder were treated with corticosteroid injections. A more recent study also examined whether the type of massive rotator cuff tear was associated with the outcome from physiotherapy, taking into account what they term “confounders” such as body mass index, length of symptoms and tobacco use [57]. This is the first published study in patients with MIRCT that used multivariate analysis to investigate the influence of lifestyle factors on functional outcomes following physiotherapy. The authors were unable to replicate the findings of Collin et al.’s study and did not find an association between tear of subscapularis and lack of success with physiotherapy. Rather, they found an association between length of symptoms, body mass index and tobacco consumption and outcomes. These results should, however, be interpreted with caution as the study was small (*n* = 92), did not follow PROGRESS guidelines [58] for prognostic studies and more specifically, did not include a no-treatment group.

Interestingly, a randomized placebo-controlled trial also concluded that teres minor hypertrophy was an essential factor in response to an anterior deltoid exercise program [37]. Of note, the study defined a massive tear as a tear greater than 5 cm and did not outline clear criteria for deeming the tear irreparable. The trial demonstrated significant improvements in pain and function in patients with massive rotator cuff tears following participation in the exercise program. These improvements were significant at three and six months, but by twelve months, although the gains were sustained, they were no longer statistically significant compared to the control group. The authors proposed that patients with improved teres minor recruitment would have a better outcome. However, the study’s design did not include a method for testing this hypothesis, and they did not measure or report increases in teres minor size or strength.

A retrospective non-randomized study of patients with MIRCT (*n* = 108) also attempted to identify important factors moderating response to physiotherapy [59]. The patients underwent either non-operative management, arthroscopic debridement, or rotator cuff repair and were evaluated using a shoulder rating questionnaire. The authors reported that up to 75% of patients who underwent non-operative management had excellent or good outcomes. They concluded that the factors that predicted a negative response to any of the treatment approaches were the presence of glenohumeral arthritis, decreased passive range of motion, superior migration of the humeral head, the presence of muscle atrophy, and weakness of external rotation or abduction strength. However, we cannot have high confidence in these conclusions as the study methodology was not designed to evaluate predictors of treatment response and did not follow PROGRESS guidelines for prognostic studies. Table 2 summarises the studies identifying possible predictors of response to physiotherapy in patients with MIRCT.

## 6. Discussion

Our literature review highlighted that the results for patients undergoing physiotherapy for MIRCT are highly variable. Even studies with almost identical treatment approaches and similar cohorts of patients demonstrated very different results. All studies were low quality, and there were no randomized controlled trials. We also found a lack of consensus on the best physiotherapy program. Finally, we identified several factors as *possibly* being related to whether patients benefit from physiotherapy. Active range of motion of less than 50° flexion, complete tear of subscapularis, lack of teres minor hypertrophy, glenohumeral arthritis, passive restriction of movement, and weakness of external rotation or abduction strength are all plausible biomechanical factors that may be related to an unsuccessful response to physiotherapy. However, it is not clear if these genuinely explain the variability of outcome success from physiotherapy in MIRCT.

### 6.1. Distinguishing between Predictors of Treatment Effect and Prognostic Factors

Firstly, when we are evaluating the literature in this area, it is essential to distinguish predictors of treatment effect from prognostic factors. Prognostic factors are patient characteristics that identify subgroups of patients that have different outcomes *without* or regardless of treatment. Predictors of treatment effect are characteristics that identify subgroups of patients having different outcomes in response to treatment [60]. A predictor of treatment effect can only be assessed using a valid comparative trial—ideally a randomized controlled trial. Therefore, studies to identify predictors of treatment effect should be nested in randomized controlled trials comparing the effectiveness of the treatment in question with that of a control or different treatment [61] (p. 189). To date, all the studies that have explored predictors of response to physiotherapy in patients with MIRCT have been single-arm cohort studies [38,39,40,42,45]. Using a treatment-only cohort design, such as case series studies, means that any associations found between a factor and the outcome could be a prognostic factor and not a genuine predictor of the treatment effect.

It is important to note that hypotheses regarding potential predictors of treatment effect are often generated through single treatment arm cohort studies—non-randomized case series without comparator or no-treatment groups. This design is essential if the emphasis is to observe and explore. However, it then requires further investigation in randomized controlled trials [61] (p 193–194). As randomized controlled trials in this area seem absent, the factors identified in the literature possibly associated with response to physiotherapy for MIRCT cannot be confirmed as predictors of treatment effect. Instead, they are possible factors that need to be evaluated further in suitably designed trials.

### 6.2. An Over-Emphasis on Biomechanical Factors

Secondly, it is apparent that previous studies seeking to identify predictors of treatment effect for patients with MIRCT have placed emphasis on biomechanical factors and have not considered psychosocial factors. Indeed, if we look at other cohorts of patients with shoulder pain, psychosocial factors feature prominently in the factors most associated with outcome. For example, one high-quality longitudinal prospective prognostic study found that psychosocial factors are the strongest predictors of a successful outcome in non-operative approaches to rotator cuff-related shoulder pain [50]. More specifically, five factors demonstrated high prognostic value: baseline pain and disability, patient expectation, pain severity at rest, employment status, and pain self-efficacy. It should be noted that this study was adequately powered and adhered to the PROGRESS guidelines on methodology of prognostic studies [58].

Another high-quality multi-centre prospective cohort study—including 433 participants—of predictors of negative response to non-operative management in patients with full-thickness rotator cuff tears also reported that low patient expectation of improvement with physiotherapy predicted poor response and the need to proceed to surgery [62]. The authors concluded that from their data, a patient’s decision to have surgery is more influenced by their negative expectations of physiotherapy than by their symptoms or by biomechanical features of the rotator cuff pathology. Likewise, in a cohort of patients with large rotator cuff tears, the only variable that was predictive of outcome from physiotherapy was the patient-reported outcome measure, the RC_QOL [63]. This outcome measure includes physical symptoms and work, recreational, lifestyle, social, and emotional domains. The findings, however. should be cautiously interpreted because of the reported wide confidence intervals. Although these three studies relate to a different group of patients with shoulder pain, the psychosocial factors identified may also be significant predictive factors for response to treatment in patients with MIRCT.

There has been a growing understanding that physiotherapy needs to move towards a broader psychosocial knowledge of musculoskeletal conditions [64]. Central to this is the idea of patient-centred care. This is characterized by establishing meaningful connections, shared decision-making, self-management support, and patient-centred communication [65]. Indeed, better quality therapeutic relationships have consistently been associated with better clinical outcomes, higher patient satisfaction, and adherence to exercise programs [49,66,67,68]. Moreover, patient satisfaction has been shown to be multidimensional and includes clinical outcomes, physiotherapist features, patient features, physiotherapist–patient relationships, treatment features, and healthcare setting features [69].

In the case of patients with MIRCT, the previously suggested biomechanical factors may be important predictors of treatment effect. However, even the most ideal exercise programs are likely to fail or be undermined if therapists do not integrate patient-centred care into the programs, gain therapeutic alliance with their patients and help to create positive expectations for the outcome of treatment. The significantly different results reported by Levy et al. [39] and Yian et al. [40] following similar physiotherapy programs in similar cohorts of patients may be due to patient factors. These patient factors could be biomechanical in nature, such as the location of the rotator cuff tears, or psychosocial, such as the therapeutic alliance and the patients’ expectations of success from the program. It is possible that some patients do not improve with physiotherapy for MIRCT because of so called “ruptures” in the therapeutic relationship and the nocebo-effects of negative expectations and miscommunication [70].

Previous research on MIRCT has focused on a narrow tranche of possible predictors of treatment effect and runs the risk of making erroneous conclusions. A wide range of potential factors must be considered when evaluating response to physiotherapy treatment so that elements that could influence the outcome are not missed. An excellent example of this is the well-designed prognostic study of response to physiotherapy in shoulder pain by Chester et al. [50]. Seventy-one potential factors were evaluated including demographic information, lifestyle factors, psychosocial factors and expectations of physiotherapy, general health, shoulder symptoms, physical assessment findings, and patients’ activity levels. Identifying possible predictors for response to physiotherapy in patients with MIRCT should include factors from these domains in addition to the biomechanical factors identified in our review. Measurement of therapeutic alliance and adherence to physiotherapy should also be considered. Because of the lack of previous research in this area, one way to identify a broad range of factors to test in a prognostic cohort study is to garner expert opinion using a Delphi study methodology. Future studies aiming to identify predictors of response to physiotherapy in this patient group should follow PROGRESS guidelines for prognostic studies [58].

### 6.3. What Are the Methodological Challenges of Prognosis Studies?

The PROGRESS guidelines highlight many methodological challenges in prognosis research. However, notably when evaluating if a factor is a predictor of treatment effect, four different states must be included in a trial—negative for the marker, positive for the marker, treatment, and no treatment [58]. For example, if we were exploring the influence of having a subscapularis tear on the outcome of physiotherapy in MIRCT, then the trial would ideally need to include people with and without subscapularis tears, and each of these groups would need to have participants that underwent physiotherapy and those that had no physiotherapy. If the factor influences outcome in both people who had treatment and those who did not have treatment, then it may be a prognostic factor rather than a predictor of treatment response.

When we consider that trials should test a broad range of possible factors, few individual trials are large enough to assess whether factors are genuinely predictive of treatment effect. It is more likely that evidence will emerge over time from meta-analysis of pooled data. It would be helpful if future randomized controlled trials for response to physiotherapy in MIRCT use agreed definitions of MIRCT and expected outcomes. We suggest that MIRCT should be defined as complete tears of two or more rotator cuff tendons with Goutallier Grade 3 or 4 fatty infiltration and significant tendon retraction (Patte Grade 3). Creating a common set of outcome measures that should be used consistently across studies is a more onerous task requiring consensus agreement [71]. We suggest that one tool that deserves consideration is the RC_QOL, as it measures symptoms, work, recreational, lifestyle, social, and emotional domains.

Our literature review also revealed no consensus on what constitutes to the best physiotherapy exercise program for patients with MIRCT. There have been no randomized controlled trials comparing different physiotherapy programs with each other or with a no-treatment arm. Plausible biomechanical theories have been proposed to explain how different physiotherapy exercise approaches might help improve the centring of the head of the humerus in the glenoid and prevent superior or anterior subluxation. However, to date, no study has empirically evaluated whether these physiotherapy exercises improve the biomechanics of the glenohumeral joint. It should be a priority for future research to assess the effect of different exercises on markers, such as acromiohumeral distance in patients with MIRCT. Measurement of targeted muscles’ cross-sectional area before and after rehabilitation would also be informative. For example, this could be used to evaluate if flexion-based physiotherapy programs truly hypertrophy the anterior deltoid and whether teres minor hypertrophy occurs following rehabilitation. Given that many patients with MIRCT are asymptomatic and can gain full range of motion, it may also be informative to investigate the biomechanical characteristics of this cohort of “copers”.

It should also be a research priority to carry out randomized controlled trials comparing different physiotherapy programs with each other or with the natural history of MIRCT or with surgical intervention such as RTSA. It should also be noted that any trials that evaluate the effect of exercise therapy in this patient group should measure therapeutic alliance and ideally, incorporate ways of optimizing therapeutic alliance in the physiotherapy program. Other research priorities include carrying out a longitudinal prospective prognostic study to identify predictors of response to physiotherapy in patients with MIRCT. While the gold standard method for identifying predictors of treatment effect would be to nest this study in a randomized control trial including a no-treatment arm, we acknowledge the methodological challenges of running such a large study. Important exploratory data of predictors could be obtained from a single-arm prospective cohort study.

## 7. Conclusions and Recommendations

Our review of the literature on predictors of response to physiotherapy in patients with MIRCT revealed that the outcomes of physiotherapy in this patient group are highly variable. The reasons for this variability require further investigation. Several biomechanical factors were identified as possibly being related to a successful or unsuccessful outcome following physiotherapy, with complete tear of subscapularis demonstrating the strongest evidence. However, there were no appropriately designed prognostic studies and no randomized controlled trials comparing different physiotherapy programs, or comparing physiotherapy to natural history or surgery. There is a strong emphasis on biomechanical factors as possible predictors of treatment response. However, other domains, such as psychosocial factors and therapeutic alliance, which are important in similar patient populations, have not been explored.

### 7.1. Recommendations for Future Research

Delphi study to gain expert consensus on possible predictors of response to physiotherapy in patients with MIRCT. This qualitative work should include exploration of patients’ perspectives on this question.Randomized controlled trials to compare different physiotherapy programs for MIRCT with each other, with surgical procedures, or with a no-treatment group.Evaluation of the effect of physiotherapy exercises on their purported biomechanical aims, e.g., anterior deltoid hypertrophy, teres minor hypertrophy, and reduction of superior migration of the head of the humerus.Evaluation of the biomechanical properties of patients with MIRCT who are asymptomatic.Longitudinal prospective prognostic studies including a wide range of possible predictors of response to physiotherapy in patients with MIRCT.

### 7.2. Recommendations to Improve Clarity of Terminology

Our literature review highlighted that the terms pseudoparesis and pseudoparalysis have heterogenous definitions and interpretations and are sometimes used interchangeably. As a solution to this lack of clarity, we would suggest that it is time to move away from these terms. Firstly, the lack of agreement throughout the literature is not helpful. Secondly, the words themselves may be anxiety-provoking for patients who hear them or read them in medical reports. Finally, we can bring further clarity to the terminology around MIRCT by being more factual and precise in describing the active versus passive range of motion deficits. For example, more precise clinical information can be conveyed if we use the term “shoulder flexion lag sign” or “shoulder abduction lag sign” to describe when there is less active range of motion compared to passive range of motion. We can proceed to quantify this lag, e.g., “There is a shoulder flexion lag sign with 50° active range compared to 150° passive range”. Finally, the use of these quantifiable descriptors moves away from previous unsubstantiated assertions that pseudoparalysis and pseudoparesis represent distinctly different clinical entities or are reflective of particular tear patterns.

Similarly, our literature review highlighted a lack of evidence that “anterior deltoid programmes” preferentially activate the anterior deltoid muscle. We recommend replacing the term with the more accurate descriptor, “flexion-based rehabilitation programs”.

### 7.3. Clinical Recommendations

It is currently difficult to make clear recommendations to clinicians seeking to provide evidence-based clinical decisions about this patient group. When considering the best treatment pathway for a symptomatic patient with MIRCT, a trial of physiotherapy should be considered. Physiotherapy should involve exercises that are carried out for a minimum of twelve weeks, affect shared decision-making, and emphasize developing therapeutic alliances and interventions to raise patient expectations of the success of the treatment. Patients whose subscapularis is involved in their MIRCT may be less likely to respond to physiotherapy and could be considered for earlier surgical intervention [45].

## 8. Limitations

This review was not a systematic review with formal risk of bias assessment. This may have introduced bias in the interpretation of the literature.

## Figures and Tables

**Figure 1 ijerph-20-05242-f001:**
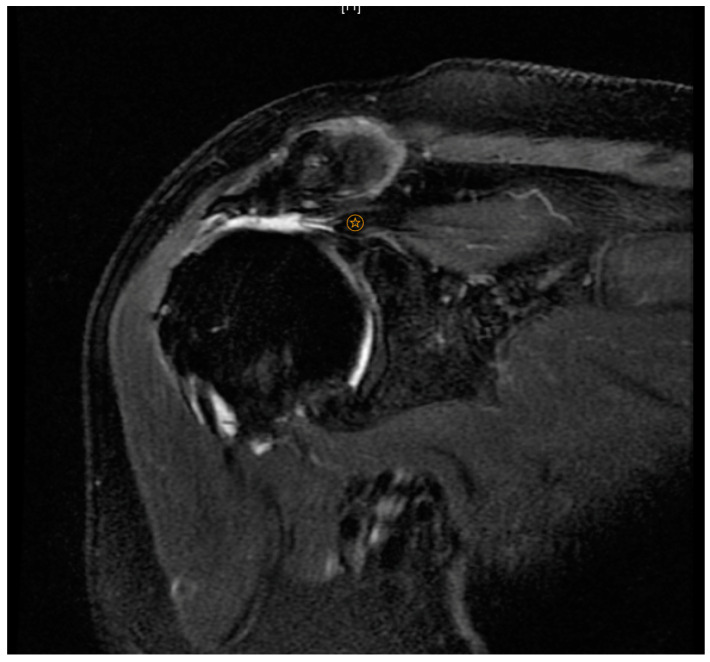
Patte Grade 3. 
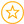
 indicates level of supraspinatus tendon retraction.

**Figure 2 ijerph-20-05242-f002:**
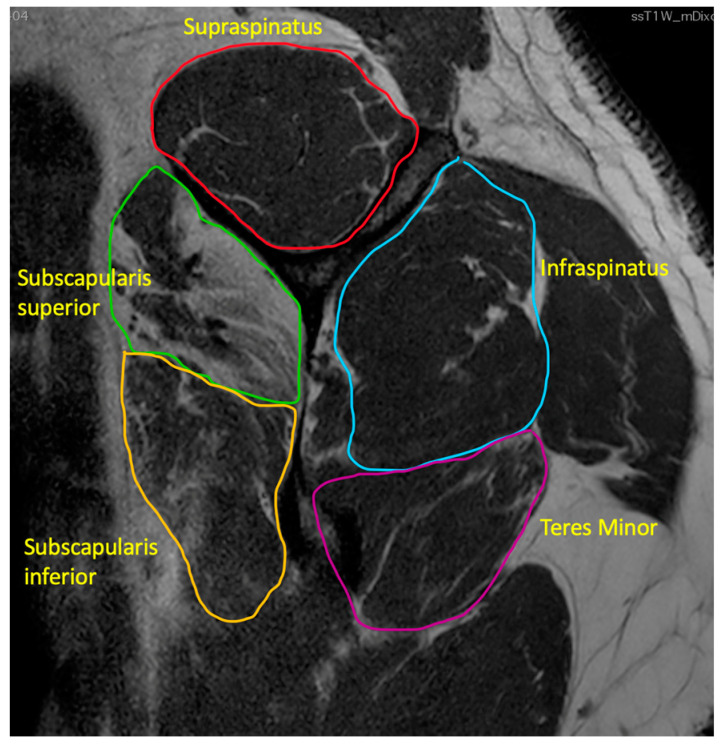
Goutallier Grade 4 subscapularis superior, Grade 3 subscapularis inferior, Grade 2 teres minor, Grade 2 infraspinatus, Grade 1 supraspinatus.

**Figure 3 ijerph-20-05242-f003:**
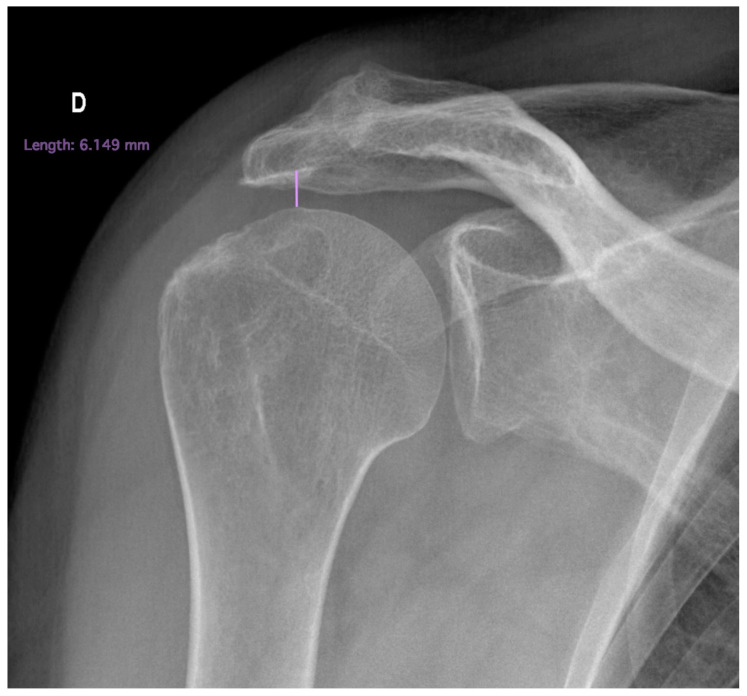
Superiorly migrated head of humerus.

**Figure 4 ijerph-20-05242-f004:**
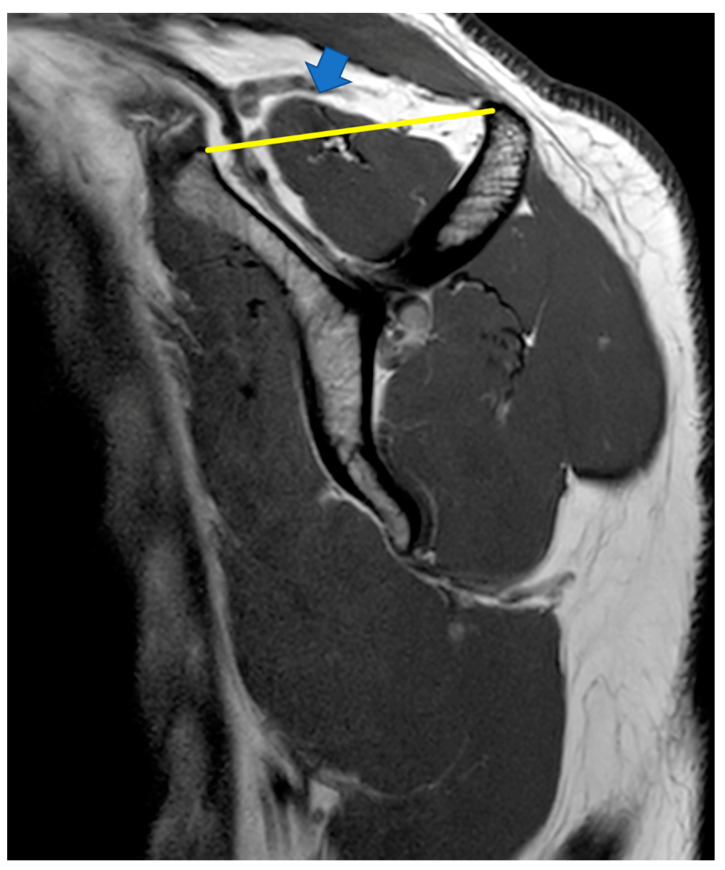
Negative tangent sign: 
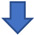
 indicates supraspinatus extending above the tangent.

**Table 1 ijerph-20-05242-t001:** Summary of studies evaluating physiotherapy for massive irreparable rotator cuff tear.

Study	Study Type	Number of Participants	Mean Age	Criteria for Defining Massive Irreparable Rotator Cuff Tear	Other Tear Types Included?	Exercise Intervention	Length of Physiotherapy Intervention	F/U Time	Definition of Successful Outcome	Definition of Failure	How Many Had Successful Outcome?	Level of Evidence
Levy et al. (2008) [39]	Prospective case series	*n* = 17	80 (range: 70–96)	Complete rupture ≥ 2 tendons, Goutallier Grade 4 fatty infiltration, Patte Grade 3 retraction, AH distance ≤ 7 mm, pseudoparalysis	No	Anterior deltoid strengthening: well described	At least 12 weeks	6 and 12 weeks, 6 and 9 months	Definition of successful outcome or failure unclear, but “adequate ROM” and “no pain medication” mentioned, but not defined.		14/17 (82%)	III
Yian et al. (2017) [40]	Prospective case series	*n* = 30	74 (range: 55–89)	Complete rupture ≥ 2 tendons, Goutallier Grade 4 fatty infiltration, Patte Grade 3 retraction	No	Anterior deltoid strengthening	3 months	9 and 24 months	Pt decision to not have surgery, >20 improvement on ASES score	Pt abandoning prog due to pain or pt’s decision to have surgery or less than 20 pt improvement on the ASES score on final follow-up	40%	III
Christensen et al. (2016) [38]	Prospective case series	*n* = 24 (not including 6 dropouts)	70 (range: 49–89)	Complete rupture ≥ 2 tendons, USS evaluation of tendon retraction or MRI (fatty infiltration/retraction or arthroscopic evaluation)	No	Anterior deltoid strengthening: well described	5 months	5 months	None	None	Not known. Mean statistical improvements only reported across PROM, QOL, strength and pain	III
Ainsworth et al. (2009) [37]	RCT (prospective, placebo controlled	*n* = 60 (6 lost to f/u)	78 (range: 68–88)	FTT > 5 cm adjudged by surgeon to be irreparable	Yes	Anterior deltoid strengthening	6 treatment sessions	3,6 and 12 months	None	None	Not known, statistically significant improvement in OSS at 3 and 6 months, not significant at 12 months	I
Collin et al. (2015) [45]	Prospective case series	*n* = 45	67 (range: 56–76)	Complete rupture ≥ 2 tendons, Goutallier ≥ Grade 3, pseudoparalysis	No	Scapular and entire deltoid strengthening not well described	5 physiotherapy sessions	2 years	Not explicitly stated, but suggested significant improvement in constant score/achievement of 160º forward elevation (24/45 achieved this)	Not explicitly stated, but suggested not achieving 160° forward elevation	24/45 (53%)	III
Guitierrez-Espinoza et al. (2018) [42]	Prospective case series	*n* = 92	68 (range: 63–72)	Complete rupture ≥2 tendons, Goutallier ≥Grade 3	No	Manual therapy and specific exercises—progressive scapular and GHJ control exercises (MT joint mobilizations well described, but exercises a bit vague)	12 weeks (x2 session/week)	12 weeks	Statistically significant improvement	Not explicitly stated	Not known, mean statistically significant improvements shown in Constant Score, DASH and VAS	III

USS: Ultrasound scan; DASH: Disabilities of the Arm Shoulder and Hand Questionnaire; VAS: Visual Analogue Scale; GHJ: Glenohumeral joint; ASES: American Shoulder and Elbow Surgeons Standardized Assessment Form; QOL: Quality of life; MT: Manual therapy; Pt: Patient.

**Table 2 ijerph-20-05242-t002:** Summary of studies identifying predictors of response to physiotherapy in MIRCT.

Study	Design	No. of Participants	Predictor	Predictor of Success/Failure?	Comments
Yian et al. (2017) [40]	Prospective case series	*n* = 30	<40° shoulder flexion at outset of program	Failure	Design of the study was focused on evaluating success or failure of physiotherapy rather than identifying predictors of success or failure. No control group
Collin et al. (2015) [45]	Prospective case series	*n* = 45	Tear involving subscapularis or three tendons torn. Tears isolated to posterior cuff	Failure Success	Study design focused on whether site of rotator cuff tear predicted outcome from physiotherapy, but did not use multivariate statistical analysis to consider whether other factors were involved in predicting response. No control group
Agout et al. (2018) [44]	Prospective case series	*n* = 71	No correlation between site of tendon tears and success or failure	n/a	Study design did not use multivariate statistical analysis to test whether other factors were involved in predicting response. No control group
Ainsworth et al. (2009) [37]	RCT (prospective, placebo controlled	*n* = 60 (6 lost to f/u)	Authors hypothesised that increased teres minor recruitment was a predictor	Success	Study design did not test this hypothesis as it did not evaluate teres minor hypertrophy or recruitment
Yoon et al. (2019) [56]	Prospective case series	*n* = 108	Intact subscapularis tendon and teres minor hypertrophy	Success	Study design did not use multivariate statistical analysis to test whether other factors were involved in predicting response. No control group
Araya_Quintanilla et al. (2021) [57]	Single group pre and post intervention study	*n* = 92	No correlation between presence of subscapularis tear and failure of physiotherapy Long duration of symptoms, high body mass index and tobacco use	n/a Failure	Correlation shown between these factors and outcome, but values for long duration of symptoms and high body mass index not defined. No control group
Vad et al. (2002) [59]	Retrospective non-randomized cohort study	*n* = 108	Presence of glenohumeral arthritis, decreased passive range of motion, superior migration of the humeral head, presence of muscle atrophy, and weakness of external rotation or abduction strength	Failure	Study design not appropriate to evaluate predictors of treatment effect. No control group

## Data Availability

Data sharing not applicable. No new data were created or analyzed in this study.

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
