# Peer review of "Massive Irreparable Rotator Cuff Tears: Which Patients Will Benefit from Physiotherapy Exercise Programs? A Narrative Review"

_ijerph, 2023, doi:10.3390/ijerph20075242_

Round 1

Reviewer 1 Report

First of all, thank you for your fascinating paper about an important topic, which has not yet gained its deserved attention. It was interesting for me, but I have a few suggestions, which you will find below.

Introduction: 

Line 95: explain high-quality trials – I would rephrase the sentence and mention the level of evidence or else.

Method

This is a major flaw. I would recommend a table with information (epidemiological numbers, therapy approaches, length of therapy, etc)  about the included case reports. In a review of conservative treatment, you should at least mention the different physiotherapy concepts which were used in the case series.  Maybe also a flow chart and a description of how the literature search was done. This is standard procedure for a review. Please mention inclusion and exclusion criteria.

2.2 Irreparable rotator cuff tears

Line 135: I would mention the ways to identify irreparable rotator cuff tears or change the numbers of the subpoints as it is a little bit confusing. (e.g. 2.2.1 etc.)             

line 306: please describe which physiotherapy approach was used, unless you mention it in the table I recommended before.

Line 366: Please define some association

In general, this is a well-structured and well-written paper, about conservative therapy concepts and their success in patients with massive irreparable rotator cuff tears. Not many studies have been done yet and this review tries to summarize the current state of the literature.

After including a table with an overview of the included studies and some minor points I would recommend publishing it.

Thank you for your work

Author Response

We thank the Reviewer for all the suggested corrections. You can find attached all the corrections we did to improve the manuscript.

Reviewer 2 Report

Dear Authors

Thanks a lot for the opportunity you have offered me to revise the fascinating manuscript "Massive irreparable rotator cuff tears: which patients will benefit from physiotherapy exercise programs? A narrative review". 

As a significant strength, this manuscript identifies possible prognostic factors for successful and unsuccessful physiotherapy outcomes in patients with MIRCT. This proposal is a novelty in the field and adds information to the existing evidence in the literature produced in the field.

As a major weakness, the manuscript sometimes needs more details and clarity concerning methodological steps that would help improve the understanding of the manuscript. Therefore, I have suggested some strategies to improve authors' reporting and increase the quality of their work. 

Overall, my peer review is a minor revision.

¶MAJOR ISSUES:

#INTRODUCTION

*framework for shoulder pain. In 2018, Ristori (doi: 10.1186/s40945-018-0050-3) proposed a framework for shoulder pain, including also the massive irreparable rotator cuff tears as specific conditions that should be managed with a physician (e.g., orthopaedic). Adding this element in the introduction could ameliorate the rationale of the authors' narrative review.

*clinical reasoning and screening for the referral. The authors' review is well-developed and balanced. However, as a clinician, I would have expected to read a short section on the clinical reasoning that the physiotherapist should have to exclude patients with shoulder pain that might mimic massive irreparable rotator cuff tears but actually have red flags. Some examples from the literature underline the importance of this process (doi: 10.1002/msc.1628. doi: 10.3390/tomography8010032 doi: 10.1080/09593985.2021.1920077 doi: 10.3233/BMR-171069.   Doi: 10.1016/j.jbmt.2020.07.005). I suggest the authors consider this point and incorporate it into their manuscript.

#METHODS

*search strategy. Although the review is not systematic, the authors should at least report the keywords used, and the databases searched. This would improve the transparency of their reporting and raise the methodological quality of the review.

#RESULTS:

* 3. What constitutes the best physiotherapy programme?: The authors refer to manual therapy techniques. These were either joint based (e.g., mobilisation, manipulation) or soft-tissue-based (e.g., trigger point). Please be more specific.

*approach to exercise: There is evidence in the literature that exercises in shoulder pain can be pain-free (e.g., contingent symptom approach - see Lewis J doi: 10.2519/jospt.2015.5941) or performed in the presence of pain (e.g., time contingent approach - see Littlewood doi: 10.1136/bjsports-2018-099063, doi: 10.1016/j.math.2013.07.005). I invite the authors to present state of the art in the presence of massive irreparable rotator cuff tears. Furthermore, what clinical criteria could guide clinicians in using exercises with or without pain considering the different pain mechanisms (e.g., nociceptive, central sensitisation)? A synoptic table might help.

*5. What are the predictors of a successful or unsuccessful response to physiotherapy 311 in patients with MIRCT?: I suggest authors to summarise in a table the variables that predict positive and negative outcomes with evidence. This strategy will elevate the paper.

*PROGRESS guideline: please, add the appropriate reference.

#DISCUSSION

* 6.3. An over-emphasis on biomechanical factors. The authors refer to over-emphasis on biomechanical factors and the importance of psycho-social aspects. This point is very good. The literature in the musculoskeletal field suggests the importance of considering these aspects (e.g., perception of experience, satisfaction with care) within the clinical reasoning process (doi: 10.1080/09638288.2018.1501102). I invite the authors to discuss this reference in the context of massive irreparable rotator cuff tears.

* patient-centred care. The authors emphasise the importance of establishing meaningful connections, shared decision-making, self-management support, and patient-centred communication. I totally agree with the relevance of this positive element. However, as clinicians, we know that very often, we are faced with more complex and sometimes negative communication situations (e.g., unmet expectations, unclear communication) that can generate nocebo effects (doi: 10.3389/fpsyg.2022.789377. doi: 10.2519/jospt.2022.11152). Given the relevance of these variables, I suggest the authors consider these references and include massive irreparable rotator cuff tears in the discussion.

*6.4. What are the methodological challenges of prognosis studies?: this section is wonderful. Congratulation!

*future suggestions: I suggest adding a statement on the importance of assessing the patients' perspective using a qualitative study design (e.g., focus group, interview).

*Limitations. I suggest repeating as a limitation the lack of systematic research that may have inserted bias in the interpretation of the literature. This approach is fairer and increases the manuscript's transparency in the reader's eyes.

¶MINOR ISSUES:

#ENGLISH:

*need for manuscript revision: I suggest the authors have the English revised by a native speaker. In fact, there are several typos and inaccuracies (e.g., points and grammatical errors). 

#FIGURE

*Figure 1. The star is not visible in yellow. Please, ameliorate the quality of the figure.

*Figure 2. The circle are not visible. Please, ameliorate the quality of the figure.

*Figure 4. The arrow is not visible. Please, ameliorate the quality of the figure.

#REFERENCES:

*date: many references are not recent. I suggest including more up-to-date refs on the topic (e.g. from the last 2-3 years).

#TABLES

*Acronyms. Report in full all the acronyms presented in all tables. 

Author Response

(The authors gave the same response as above.)
